# Clinical Utility of Next-Generation Sequencing-Based Panel Testing under the Universal Health-Care System in Japan: A Retrospective Analysis at a Single University Hospital

**DOI:** 10.3390/cancers13051121

**Published:** 2021-03-05

**Authors:** Chiaki Inagaki, Daichi Maeda, Kazue Hatake, Yuki Sato, Kae Hashimoto, Daisuke Sakai, Shinichi Yachida, Norio Nonomura, Taroh Satoh

**Affiliations:** 1Department of Frontier Science for Cancer and Chemotherapy, Graduate School of Medicine, Osaka University, Osaka 565-0871, Japan; cinagaki@cfs.med.osaka-u.ac.jp; 2Department of Clinical Genomics, Graduate School of Medicine, Osaka University, Osaka 565-0871, Japan; daichimaeda@cgp.med.osaka-u.ac.jp; 3Center for Cancer Genomics and Personalized Medicine, Osaka University Hospital, Osaka 565-0871, Japan; k-hatake@mccg.med.osaka-u.ac.jp (K.H.); dsakai@cfs.med.osaka-u.ac.jp (D.S.); syachida@cgi.med.osaka-u.ac.jp (S.Y.); nono@uro.med.osaka-u.ac.jp (N.N.); 4Department of Genetic Counseling, Osaka University Hospital, Osaka 565-0871, Japan; ysato@hp-gensel.med.osaka-u.ac.jp (Y.S.); kae.h@gyne.med.osaka-u.ac.jp (K.H.); 5Department of Obstetrics and Gynecology, Graduate School of Medicine, Osaka University, Osaka 565-0871, Japan; 6Department of Cancer Genome Informatics, Graduate School of Medicine, Osaka University, Osaka 565-0871, Japan; 7Department of Urology, Graduate School of Medicine, Osaka University, Osaka 565-0871, Japan

**Keywords:** next-generation sequencing, solid cancer, universal health-care system, precision medicine, presumed germline findings

## Abstract

**Simple Summary:**

Next-generation sequencing (NGS)-based assay is widely used in clinical practice due to its reimbursement by Japan’s universal health-care system for cancer patients who finished standard treatment in June 2019. To clarify the clinical utility of the NGS assay under the universal health-care system, we retrospectively analyzed patients who underwent NGS assay at our hospital. Since reimbursement of the NGS assay is restricted to patients who complete standard treatment, many patients experience clinical disease progression before receiving results; therefore, they could not use the NGS results for making a therapeutic decision. Broader reimbursement of NGS assays for advanced cancer patients is needed for making optimum use of the NGS assay results. Providing good access to clinical trials and off-label agents is necessary for enabling patients to benefit from NGS assay. Additionally, this study revealed that the disclosure of presumed germline findings is feasible in clinical practice.

**Abstract:**

Next-generation sequencing (NGS) assay is part of routine care in Japan owing to its reimbursement by Japan’s universal health-care system; however, reimbursement is limited to patients who finished standard treatment. We retrospectively investigated 221 patients who underwent Foundation One CDX (F1CDx) at our hospital. Every F1CDx result was assessed at the molecular tumor board (MTB) for treatment recommendation. Based on patients’ preferences, presumed germline findings were also assessed at the MTB and disclosed at the clinic. In total, 204 patients underwent F1CDx and 195 patients completed the analysis; however, 13.8% of them could not receive the report due to disease progression. Among 168 patients who received the results, 41.6% had at least one actionable alteration, and 3.6% received genomically matched treatment. Presumed germline findings were nominated in 24 patients, and 16.7% of them contacted a geneticist counselor. The NGS assay should be performed earlier in the clinical course to maximize the clinical benefit. Broader reimbursement for the NGS assay would enhance the delivery of precision oncology to patients. Access to clinical trials affects the number of patients who benefit from NGS. Additionally, the disclosure of presumed germline findings is feasible in clinical practice.

## 1. Introduction

Over the last decade, with the increased knowledge in molecular profiles and mechanisms, there has been significant progress in cancer research and treatment. Next-generation sequencing (NGS) allows the sequencing of a large number of genes in a short time at an affordable cost and therefore contributes to detecting clinically relevant alterations and promoting precision oncology. Several studies have shown that molecular profiling with NGS improves patient response and survival in a selected cohort [1,2,3,4,5].

For example, the Molecular Screening for Cancer Treatment Optimization (MOSCATO 01) study demonstrated that targeted therapy, which was matched to a genomic alteration, improved the survival of 33% (63/193) of the study participants [4]. In addition, in the Targeted Agent and Profiling Utilization Registry (TAPUR) study, genomically matched treatment showed good clinical efficacy in the following five cohorts: pertuzumab and trastuzumab in *ERBB2*-amplified or overexpressed colorectal cancer [6], emurafenib and cobimetinib in *BRAF* V600E/D/K/-mutated colorectal cancer [7], pembrolizumab in metastatic breast cancer with a high mutational burden [8], pembrolizumab in metastatic colorectal cancer with a high mutational burden, and palbociclib in non-small cell lung cancer with *CDKN2A* alteration [9,10].

NGS assay is widely considered a part of the routine care for patients with cancer, and it has been reimbursed in several Western and Asian countries [11]. In June 2019, two types of NGS-based panel testing, Foundation One CDX (F1CDx, developed by Foundation Medicine, Cambridge, MA) and OncoGuide NCC Oncopanel System test (developed by Japan’s National Cancer Center; NCC and Sysmex), were reimbursed by Japan’s universal health insurance system for patients with advanced cancer who finished standard treatment [12,13,14]. Although this approval is a big step for advancing precision oncology in Japan, its application is still challenging due to the complexity of the interpretation of genetic profiles and integration of personalized treatment into the health-care system. To investigate the clinical utility of NGS in daily practice, we reviewed patients who underwent F1CDx assay under the universal health-care system at our hospital. Herein, we present precise data of the patient characteristics, genetic alterations, including presumed germline variants nominated by the molecular tumor board (MTB) and subsequent treatment.

## 2. Results

### 2.1. Feasibility of Next-Generation Sequencing (NGS) Assay and Patient Characteristics

Samples were received from 213/221 patients, and nine were withdrawn following a pathologist evaluation on tumor volume in the samples (Appendix A). A total of 204 were assayed with F1CDx, and 195 samples (95.6%) were successfully analyzed. Reasons for analysis failure were insufficient tumor volume (n = 4), insufficient DNA quality (n = 4), and contamination (n = 1). A total of 168 (86.6%) patients received their F1CDx results and MTB-approved report at the clinic, while 27 (13.8%) could not due to disease progression (death; n = 10, declining conditions; n = 17). The median turnaround time, which is defined as the duration between the date of sample reception and the date of the MTB, was 43 days (range 35–51 days).

The patient and disease characteristics of 168 patients are listed in Table 1. The median age of the patients was 62 (range 3–92) years, and 163 (97%) patients had an Eastern Cooperative Oncology Group performance status (ECOG PS) of 0–1, while five (3%) patients had an ECOG PS of 2. Most of the patients were heavily pre-treated, and the median number of previous chemotherapy lines was 3 (range 1–11). Nearly half of the patients (n = 75, 44.6%) were referred for NGS from smaller partner community-based hospitals in the region. The most frequent tumor types were colorectal cancer (n = 45, 26.8%), sarcoma (n = 22, 13.1%), and pancreatic cancer (n = 18, 10.7%). The median survival time was 217 days (95% confidence interval; 95%CI 185–262 days).

A summary of the genetic alterations is shown in Figure 1A. The median number of genetic alterations per tumor was 4.72 (range 0–14). The median tumor mutational burden (TMB) was 2.52 (range 0–21.42), and eight patients had TMB–high (TMB-H) (Figure 1B).

### 2.2. Matched Treatment According to Actionable Mutation

Among the 168 patients who received their results, 107 actionable alterations were found in 70 (41.6%) patients (Figure 2, Appendix A). The median number of actionable mutations per person was 1.53 (range 1–5). The frequencies of each OncoKB level of evidence were as follows: level 1A, 8.4% (n = 9); level 2, 5.6% (n = 6); level 3A, 5.6% (n = 6); level 3B, 41.1% (n = 44); and level 4, 15.9% (n = 17). The most frequently annotated genes were *PIK3CA* (n = 18), *TP53* (n = 11), *ERBB2* (n = 9), *MDM2* (n = 6), and *FGFR3* (n = 3) (Appendix A). One patient had a recommendation of off-label treatment only, 13 patients had a recommendation of off-label treatment and clinical trials, and 56 patients had a recommendation of clinical trials. Additionally, 14 patients had a recommendation of mutation-driven clinical trials that were ongoing at our institution. Based on the MTB recommendation, six (3.6%) patients were treated with targeted treatment (Figure 2, Appendix A). Four patients were enrolled in five genomically matched clinical trials, four of which were conducted at our institution. Two patients used targeted agents in the off-label treatment, and it was beneficial to one of them (Figure 3). She was a 75-year-old female patient with pre-treated metastatic cholangiocarcinoma harboring an *ERBB2* amplification (Copy number; CN = 114) and treated with dual human epidermal growth factor receptor 2 (HER2) blockage therapy (trastuzumab and pertuzumab), and a good clinical response was observed for 9 months until the appearance of pleural effusion. Following pleural adhesion, the next treatment was initiated with trastuzumab deruxtecan, an HER2-targeting antibody–drug conjugate. She achieved tumor shrinkage after 1.5 months of the treatment, but she requested treatment discontinuation due to grade 3 fatigue, which gradually subsided several weeks following the discontinuation.

### 2.3. Presumed Germline Findings

A total of 166 (98.8%) patients preferred to be informed about the presumed germline findings, and 156 (95.1%) adult patients wanted to share the findings with their family members (Appendix A). A total of 26 presumed germline pathogenic variants in 24 patients (14.3%) were nominated by a germline-focused tumor analysis in the following genes: *SMAD4* (n = 6), *BRCA2* (n = 4), *PTEN* (n = 3), *BRCA1* (n = 3), *RB1* (n = 2), *STK11* (n = 2), *ATM* (n = 1), *BRIP1* (n = 1), *MSH6* (n = 1), *RAD51* (n = 1), *TP53* (n = 1), and *TSC2* (n = 1) (Table 2). All the findings were described in the MTB-approved report and returned to the patients. Five of them (20.8%) contacted a genetic counselor, and one patient proceeded for further germline testing.

## 3. Discussion

This study presented the real-world data of patients with advanced malignancies who exhausted their standard treatment and underwent NGS at our institution. The NGS assay had a good feasibility in clinical practice with a high success rate and an ordinary turnaround time [15]. MTB recommendations, subsequent genomic-matched treatment, and management of presumed germline findings in daily practice were also presented. Genes that recurrently altered across samples and the percentage of patients who were provided MTB recommendation were similar to that in other series; however, the number of patients who received a targeted agent based on the NGS findings in our cohort is smaller than that in previous reports [16,17,18,19]. There are several explanations for the low rate of treatment with the genomically matched drug received in this study.

First of all, the timing for NGS assay appeared to be too late for making optimum use of its results. Under the Japanese universal health-care system, reimbursement of the NGS assay is restricted to patients who have completed their standard treatment and are eligible for palliative treatment. As a result, we found twenty-seven patients (27/204, 13.2%) who experienced disease aggravation or death during the wait for NGS results; the NGS results were not considered for therapeutic decision making. In addition, disease progression is a major limiting factor for the initiation of treatment after NGS assay, as described in previous literature [19]. The optimal timing for NGS assay in patients with cancer has not yet been determined. However, our study suggested that to obtain the maximum therapeutic value of NGS, it should be performed early in the course of the disease. A prospective study on the feasibility and utility of large NGS assays before initial systemic treatment is ongoing, with the aim of reimbursement of NGS assays in the frontline setting for metastatic cancer patients in Japan [20]. Nearly half of the patients who underwent NGS assay were referred from smaller partner community-based hospitals that do not have MTB. To make the best use of NGS, physicians and medical staff need to be encouraged to consider early referral for panel test assessment.

Secondly, limited access to early phase clinical trials is a major barrier for enrolling patients in matched clinical trials, as mentioned in previous articles [17,21]. A recent report from National Cancer Center Hospital (NCCH) demonstrated that 13.3% (25/230) of the patients who underwent NGS after completing their standard chemotherapy were treated with matched targeted agents based on the MTB recommendation; this rate is approximately four times higher than that of our cohort (6/168, 3.6%) [16]. NCCH is a leading facility in early phase drug development in Japan, and it runs the largest number of early phase clinical trials [22,23]. Therefore, they have a greater opportunity for the patients to be enrolled in genomic-driven trials of a drug in development. This leads to a disparity in the number of patients who received matched targeted agents between the hospitals. A new basket/umbrella trial, which is similar to the TAPUR study, was started at our institution in July 2019. It provides 15 targeted agents that were reimbursed in other indications for patients with matched actionable mutations [24]. This trial would partially improve access to targeted therapy. Additionally, consultation via a virtual platform is gradually being adapted in oncology [25,26]. The integration of telemedicine in clinical trials to enhance clinical trial accessibility is anticipated.

Thirdly, it is difficult to access investigational targeted agents outside the clinical trial under the Japanese health-care system. We do not have a system similar to the expanded access program in the United States and Europe. In addition, all the costs related to off-label use generally need to be paid out-of-pocket, and very few patients can afford it. Moreover, each case must be approved by an institutional review committee before prescribing an off-label treatment [27]. Such circumstances make physicians recommend strict off-label use. Consequently, our MTB recommended off-label use in 8.3% (14/168) of the patients in this study, and one of two patients who received off-label treatment had a favorable clinical outcome. Our MTB recommended off-label use for the genetic alterations that responded beneficially to matched treatment in previous clinical trials and case series such as *ERBB2*, *BRCA1*, *BRCA2*, and *BRAF* V600E. The clinical benefit and potential side effects of off-label use are controversial. Previous reports revealed that off-label use without concrete clinical evidence could be harmful to the patients [28,29]. If the indications for off-label use by the MTB are increased, it may increase the number of patients who use off-label agents; however, it is unlikely to be beneficial to several patients. Therefore, we believe that our conservative approach toward off-label use is reasonable in current practice.

The management of presumed germline findings is of increasing importance. A recent recommendation from the European Society of Medical Oncology (ESMO) advocates for the active disclosure of presumed germline findings upon tumor-only sequencing. In addition, the American College of Medical Genetics (ACMG) recommends the reporting of presumed germline findings, even when those found in the genes are unrelated to the primary medical reason for genome sequencing. We found that most of the patients in this study provided consent for reporting presumed germline findings to themselves and their family members. This has been addressed by several Western studies [30,31]. We understand that Japanese and Western patients have similar preferences for presumed germline findings. While 26 presumed germline findings were nominated in 24 individuals, five patients contacted genetic counselors, and one of them underwent further investigation. We learned that our management of presumed germline findings is practically acceptable in the current health-care system. The presence of a genetic specialist is not mandatory when returning presumed germline findings to patients; however, compared to a previous report, this may result in a small number of patients accessing further genetic consultation and testing [32]. We should reconsider and improve our approach for returning presumed germline findings in cooperation with cancer genetic specialists.

This study had several limitations. This was a single-canter, retrospective study. The patient population was heterogeneous, and several patients with extremely advanced disease who waited for approval of the assay were included. Given a short follow-up period of 6 months, the certain number of patients lost to follow-up, and the small number of patients who received targeted treatment, the survival analyses are not statistically reliable and thus are not shown. The presumed germline findings nominated in this study are based on the germline-focused analysis of tumor-only sequencing panel. Therefore, the clear distinction between somatic and germline mutations is difficult, and the interpretation of the findings needs careful consideration.

The strength of the study is that we presented the first real-world data of patients with various cancers who underwent NGS under the universal health-care system.

## 4. Materials and Methods

### 4.1. Patients

We retrospectively reviewed the medical records of 221 consecutive patients at Osaka University, who provided their consent to take the F1 CDx *covered* by the Japanese public health *insurance* system from September 2019 to July 2020. The median follow-up period was 179 days (range: 48–439 days). The patients’ clinical data were extracted from their medical records.

### 4.2. The Flow of NGS Assay under National Health Insurance Coverage in Japan

Details of the workflow of the NGS assay under Japan’s universal health care are found in previous studies [30,31,32]. Briefly, patients with a histopathological diagnosis of a solid tumor who finished or have finished their standard chemotherapy were candidates for insurance-covered NGS. Patients aged below 20 years provided their assent, while consent was obtained from their parents/guardians with patients’ assent. When consent was obtained, patients (and parents/guardians) were also asked whether they wanted to be informed of the results of the presumed germline variants by the physicians (see Section 4.4). Archival formalin-fixed paraffin-embedded (FFPE) tumor samples (or 20 serial unstained slides) were collected and pre-screened by board-certified pathologists at Osaka University to estimate the duration of storage and tumor content of the specimen, and then, they were sent for NGS assay (F1 CDx), which was carried out following the previously described manufacturer’s (Foundation Medicine) instructions [33,34]. Concisely, F1CDx detects 324 genes, including all coding exons of 309 cancer-related genes, one promoter region, one noncoding RNA, and select intronic regions of 34 commonly rearranged genes, the coding exons of 21 of which are also included. F1CDx also simultaneously profiled for TMB as well as microsatellite instability (MSI) status. We sent thin-sectioned FFPE slides to a Clinical Laboratory Improvement Amendments (CLIA)-certified and College of American Pathologists (CAP)-accredited laboratory (Foundation Medicine, Cambridge, MA, USA). After the pathology review of the specimen, DNA was extracted and quantified prior to Library Construction (LC). Libraries passed the quality control were hybridized and then sequenced. Sequence data were analyzed using proprietary software developed by Foundation Medicine, and quality control criteria that included tumor purity, DNA sample size, tissue sample size, library construction size, and hybrid capture yields were employed. Sequence data were mapped to the human genome (hg19) using BWA v0.5.9 [35], PCR duplicate reads were removed, and sequence metrics were collected using Picard 1.47 [36] and SAMtools 0.1.12a [37]. Local alignment optimization was performed using GATK 1.0.4705 [38]. Variant calling was performed only in genomic regions targeted by the test. TMB was measured by counting all coding synonymous and nonsynonymous (SNVs) and indels present at ≥5% allele frequency and filtering out potential germline variants according to published databases of known germline polymorphisms, including Single Nucleotide Polymorphism Database (dbSNP) and Exome Aggregation Consortium (ExAC). MSI status was determined by analyzing 95 intronic homopolymer repeat loci (10–20 bp long in the human reference genome) with adequate coverage on the F1CDx assay for length variability and compiled into an overall MSI score via principal components analysis (PCA). The report of the F1CDx as well as variant call file were assessed for the actionability of each alteration by consulting databases, such as ClinVar, Catalogue of Somatic Mutations in Cancer (COSMIC), and availability of genomically matched clinical trials and off-label agents in Japan at our own MTB with primary care clinicians, clinical oncologists, genomic counselor, clinical geneticists, and pathologists, which is a mandatory procedure under the universal health-care system. Subsequently, the MTB-approved report for the assay with the treatment recommendation was provided. The report was returned to the patient and/or their family from his/her primary clinician at the clinic.

### 4.3. Identification and Classification of Genes with Treatment Recommendation

We defined actionable mutations as mutations for whom genomically matched treatment was recommended by the MTB-approved report. Oncogenic alterations revealed by the previous testing were excluded unless genomically matched therapies beyond the standard of care were available. Genetic alterations that predicted resistance to a targeted agent were also excluded. Additionally, MTB recommendation on the TMB underwent a shift during the study period, reflecting the Food and Drug Administration (FDA)’s approval of pembrolizumab for the treatment of adult and pediatric patients with unresectable or metastatic TMB-H (≥10 mutations/megabase (mut/Mb)) solid tumors. All actionable mutations were classified according to the OncoKB levels of evidence classification as follows [39]: level 1, FDA-approved biomarker predictive of response to an FDA-approved drug in a specific cancer type; level 2A, standard care biomarkers of response to an FDA-approved drug in a specific indication; level 2B, standard care biomarkers predictive of response to an FDA-approved drug in another indication; level 3A, compelling clinical evidence in reported tumor types, which were regarded as biomarkers of therapeutic response for novel targeted agents that are not yet approved in the standard of care; level 3B, compelling clinical evidence reported in other tumor types, which are regarded as the biomarkers of therapeutic response for novel targeted agents that are not yet approved in the standard of care; and level 4; non–FDA-recognized biomarkers that are predictive of response to novel targeted agents based on compelling biologic data.

### 4.4. Presumed Germline Findings

A germline-focused tumor analysis was carried out after consent was obtained. We assessed the presumed germline findings following the proposal of the Japan Agency for Medical Research and Development (AMED) study group concerning the information transmission process in genomic medicine [40] (Appendix A). Briefly, a clinical genetic expert (K.H) extracted the data of 43 genes that were recommended for a presumed germline finding analysis (Appendix A) from a variant call format file and investigated the variant classification by consulting databases, such as ClinVar and COSMIC. A certified genetic counselor (Y.S) and a clinical geneticist (K.H) assessed the clinical utility of each alteration in terms of allele frequency and correlation to patient and family history as well as clinical findings. For *BRCA1* and *BRCA2*, pathological and likely pathogenic alterations were disclosed as presumed germline findings irrespective of allele frequency of the variants. Pathological and likely pathogenic mutations found in other genes were generally disclosed to be presumed germline variants when the variant allele frequency was ≥30% for single nucleotide substitutions and ≥20% for small insertions/deletions. Regarding *APC, RB1, TP53,* and genes of which variant of allele frequency is lower than the threshold described above, the patients’ phenotypes were carefully evaluated before disclosure. The assessment was shared and discussed as a way of disclosure at the MTB. Results of the presumed germline findings assessment were described in the MTB-approved report and were returned to the patient and/or their family by his/her primary clinician. Genetic counseling and confirmatory testing are offered to the patient when presumed germline finding is disclosed.

### 4.5. Statistical Analysis

Statistical analyses were performed using EZR (Saitama Medical Center, Jichi Medical University, Saitama, Japan), which is a graphical user interface for R (The R Foundation for Statistical Computing, Vienna, Austria). Most of our data are descriptive. The Kaplan–Meier method was used to estimate overall survival rates. The study was conducted in accordance with the Declaration of Helsinki and Good Clinical Practice guidelines. The study was approved by the Osaka University Institutional Review Board, and all patients provided written informed consent for the use of their genomic and clinical data for research purposes.

## 5. Conclusions

In conclusion, this article highlighted the current status and problems of the clinical utility of NGS assay under the universal health coverage system at a single university hospital in Japan. Though it is a top priority in precision oncology to match patients with the appropriate treatment or clinical trials, a small number of patients received genomically matched treatment based on the NGS results. Reimbursement of NGS in the universal health-care system in Japan is restricted to patients who completed their standard treatment, and quite a few patients experience disease progression before they receive their results. This led to a decrease in the number of patients whose results could be used to guide treatment decision-making and administration of matched targeted treatment. NGS assay should be considered earlier in the course of the disease to maximize the therapeutic opportunities after testing. We eagerly hope that NGS reimbursement is done for advanced cancer patients earlier in the course of the disease. The availability of clinical trials in the region is a barrier to patients benefiting from NGS. Our study demonstrated the feasibility of managing presumed germline findings in daily practice. NGS would help bring personalized cancer medicine to routine clinical practice. Adequate integration of NGS in the health-care system is required to promote the efficient clinical application of NGS and advance precision medicine.

## Figures and Tables

**Figure 1 cancers-13-01121-f001:**
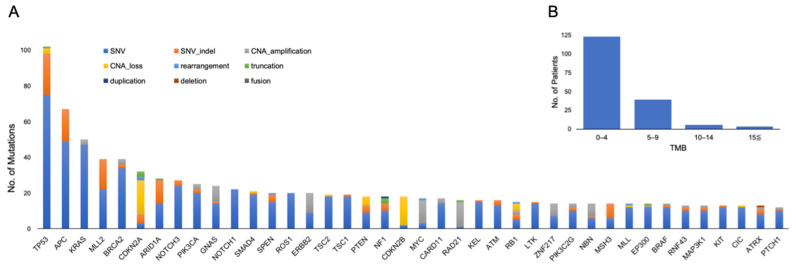
(**A**) Top 40 genomic alterations and (**B**) distribution of tumor mutational burden (TMB) in 168 patients who completed analysis. CAN: copy number alteration; SNV: single nucleotide variant.

**Figure 2 cancers-13-01121-f002:**
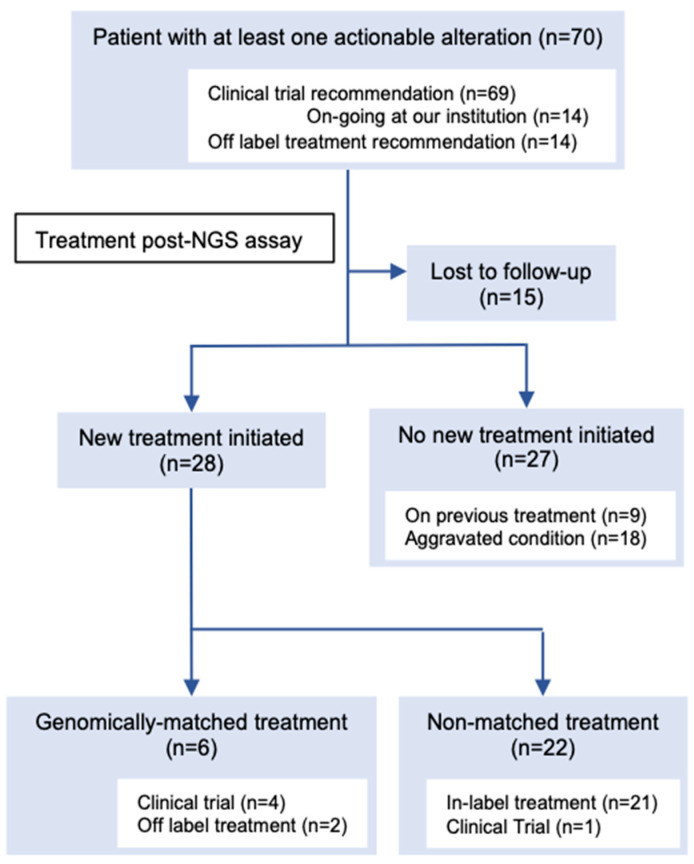
Consort diagram of post-next-generation sequencing (NGS) treatment of patients with at least one actionable alteration on the molecular tumor board (MTB) report. All numbers do not add up because some patients were counted in more than one category (i.e., had an actionable alteration with recommendations of clinical trials and off-label treatment). See Appendix A for detailed information on the patients who received genomically matched treatment.

**Figure 3 cancers-13-01121-f003:**
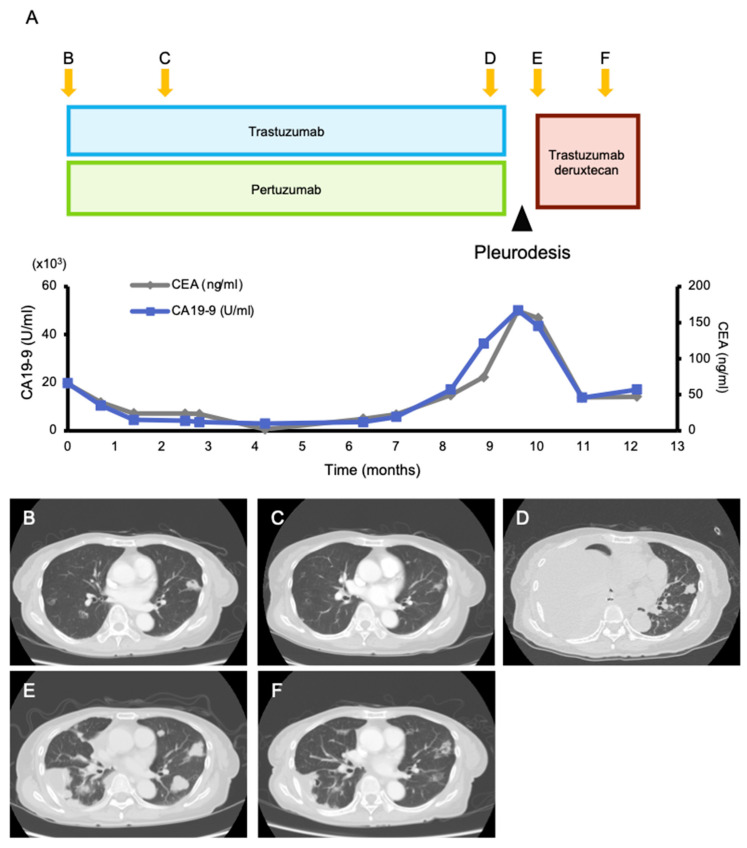
Clinical presentation. (**A**) The course of tumor markers (carcinoembryonic antigen (CEA) and carbohydrate antigen 19-9 (CA 19-9)) and (**B**–**F**) contrast-enhanced computed tomography (CT) images while receiving treatment with trastuzumab/pertuzumab and trastuzumab deruxtecan. Multiple lung metastases and liver metastases (which is not shown here) were observed when treatment with trastuzumab/pertuzumab was initiated (**B**). Two months after, a good partial response was obtained (**C**). After 9 months of treatment, the tumor became refractory to trastuzumab/pertuzumab, and a massive right pleural effusion was developed (**D**). After improvement of the pleural effusion with pleurodesis (**E**), the next treatment with trastuzumab deruxtecan was initiated, and tumor shrinkage was observed 1.5 months later (**F**).

**Table 1 cancers-13-01121-t001:** Patient demographics and characteristics.

Total		168
Sex	Male (n, %)	87 (51.8)
	Female (n, %)	81 (48.2)
Age	Median (min/max)	62 (3/92)
ECOG PS	0 (n, %)	131 (78.0)
	1 (n, %)	32 (19.0)
	2 (n, %)	5 (3.0)
No. of previous chemotherapy lines	Median (min/max)	3 (1/11)
Referral to our hospital for NGS assay	Yes, n (%)	75(44.6)
	No, n (%)	93(55.4)
Tissue of Origin	Primary site (n, %)	111 (66.0)
	Metastatic site (n, %)	57 (34.0)
Turnaround Time	Average (min/max)	43 (35/51)
Cancer Type	Colorectal	45 (26.8)
	Sarcoma	22 (13.0)
	Pancreatic	18 (10.7)
	Gastric	13 (7.7)
	Ovarian	11 (6.5)
	Bile duct	9 (5.4)
	Esophageal	8 (4.8)
	Breast	7 (4.2)
	Cervical	6 (3.6)
	Small Intestinal	5 (3.0)
	Hepatocellular	3 (1.8)
	Unknown Primary	3 (1.8)
	Endometrial	3 (1.8)
	Non-Small Cell Lung	3 (1.8)
	Brain	3 (1.8)
	Neuroblastoma	3 (1.8)
	Melanoma	3 (1.8)
	Kidney	1 (0.6)
	Prostate	1 (0.6)
	Urinary tract	1 (0.6)

NGS: next-generation sequencing; ECOG PS: Eastern Cooperative Oncology Group performance status.

**Table 2 cancers-13-01121-t002:** Presumed germline findings nominated on MTB reports.

Gene	Cancer Type	SNV Function	SNV Nucleotide Change	SNV Amino Acid Change	RefSNP Number	CNA Number of Exons	CNA Position
*ATM*	Small intestinal	frameshift	c.6710dup	p.E2238fs*11	-	-	-
*BRCA1*	**Ovarian**	**nonsense**	**c.2800C > T**	**p.Q934 ***	**rs80357223**	-	-
	Small intestinal	missense	c.236T > G	p.F79C	-	-	-
	Ovarian	missense	c.5557T > A	p.Y1853N	-	-	-
*BR* *CA* *2*	Ovarian	nonsense	c.6952C > T	p.R2318*	rs80358920	-	-
	**HCC**	**frameshift**	**c.5110_5113delAGAA**	**p.R1704fs*1**	-	-	-
	**Small intestinal**	**missense**	**c.8524C > T**	**p.R2842C**	**rs80359104**	-	-
	**Pancreatic**	**nonsense**	**c.7969A > T**	**p.K2657 ***	-	-	-
*BRIP1*	Ovarian	nonsense	c.1741C > T	p.R581 *	rs780020495	-	-
*MSH6*	Esophageal	missense	c.1082G > A	p.R361H	rs63750440	-	-
*PTEN*	Uterine	nonsense	c.733C > T	p.Q245 *	rs786202918	-	-
	Ovarian	missense	c.376G > A	p.A126T	rs1554898129	-	-
	Breast	nonsense	c.295G > T	p.E99 *	-	-	-
*RAD51*	Gastric	frameshift	c.1dup	p.M1fs	rs55714242	-	-
*RB1*	Sarcoma	frameshift	c.869delA	p.N290fs*11	rs1131690901	-	-
	Colorectal (CNA_loss)	-	-	-	-	16 of 27	chr13:48881414-49010994
*SMAD4*	**Colorectal**	**missense**	**c.1487G > A**	**p.R496H**	**rs876660045**	-	-
	Colorectal	missense	c.1081C > T	p.R361C	rs80338963	-	-
	Colorectal	missense	c.290G > A	p.R97H	rs1555685159	-	-
	Colorectal	frameshift	c.282delC	p.Y95fs*15	-	-	-
	Pancreatic	nonsense	c.346C > T	p.Q116 *	-	-	-
	Bile duct	missense	c.1058A > G	p.Y353C	rs377767346	-	-
*STK11*	Gastric (CNA_loss)	-	-	-	-	8 of 9	chr19:1152647-1223171
	NSCLC	missense	c.580G > A	p.D194N	rs121913315	-	-
*TP53*	Ovarian	splice	c.672 + 1G > A	-	rs863224499	-	-
*TSC2*	Colorectal	nonsense	c.3412C > T	p.R1138 *	rs45451497	-	-

**Bold**, patients who contacted a genetic counselor; *ATM*, ataxia telangiectasia mutated; *BRCA1*, breast cancer susceptibility gene1; *BRCA2*, breast cancer susceptibility gene 2; *BRIP1*, *BRCA1* interacting protein C-terminal helicase 1; CNA, copy number alteration; HCC, hepatocellular carcinoma; *MSH6*, *mutS* homolog 6; MTB, molecular tumor board; NSCLC, non-small cell lung cancer; *PTEN*, phosphatase and tensin homolog; *RB1*, retinoblastoma 1; RefSNP, reference single nucleotide polymorphism; *SMAD4*, mothers against decapentaplegic homolog 4; SNV, single nucleotide variant; *STK11*, serine/threonine kinase 11; *TP53*, tumor protein P53; *TSC2*, tuberous sclerosis complex 2.

## Data Availability

Clinical data and all the variant data used in the conduct of the analyses are not publicly available due to the Institutional Review Board restriction in the context of protection of the privacy and confidentiality of patients in this study, but the data are possibly available if a reasonable request is made to the corresponding author.

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
