# Peer review of "Clinical Utility of Next-Generation Sequencing-Based Panel Testing under the Universal Health-Care System in Japan: A Retrospective Analysis at a Single University Hospital"

_cancers, 2021, doi:10.3390/cancers13051121_

Round 1

Reviewer 1 Report

This is a very concise, focused and well written report of the clinical utility of next-generation sequencing-based panel testing under the universal health care system in Japan. What the paper lacks in innovation it adds in usefulness for the rationale of standardised molecular genetic testing in cancer patients. The authors used a well known mutation panel (FoundationOne) widely used in the oncology field and such systematic data in different populations and health care systems particularly regarding the number of actionable mutations and matched treatments as well as germline findings are clearly needed. Important is also the frank reporting of sample turnover time, an issue that is met in most academic and non-academic institutions and where better logistic solutions are clearly needed worldwide.

Author Response

Dear Reviewer 1:

We thank the reviewer for the positive comments on our manuscript.

Reviewer 2 Report

The work submitted for review is interesting, but there aresome issues to be clarified:

  1. The methodology of the obtained results is not described in the paper. Regarding the results of the NGS technique, there are published guidelines on what information should be provided in order to publish such results in the journal. The guidelines can be found among others:

Kim J et al. Good Laboratory Standards for Clinical Next-Generation Sequencing Cancer Panel Tests. J Pathol Transl Med. 2017 May; 51(3): 191–204.Published online 2017 May10. doi: 10.4132/jptm.2017.03.14

One of the important issues is data sharing.

„The reads generated by the NGS instrument applied for the typing should be made available for re-analysis, either via deposition in a public database [e.g., the SRA, NCBI’s Genotype and Phenotype database (dbGAP) or an equivalent]”.

The authors did not provide information on how the tests were performed and where the raw results could be found.

  1. How do the authors distinguish between somatic and germinal mutations when they only evaluate the results from FFPE tumor samples?
  2. There is no information on the patients' survival time and outcome, as well as on the correlation with the found mutations.
  3. The authors incorrectly use the nomenclature concerning the notation of the names of human genes.
  4. Please include the template of the informed consent of patients, which they signed before proceeding to genetic testing.

In order to be able to draw conclusions, it is necessary to be sure that the analyzes have been carried out in accordance with applicable standards. 

Author Response

Dear Reviewer 2,

Please kindly find the file attached.

Round 2

Reviewer 2 Report

I accept explanations. I consider this data worth publishing, therefore I accept the lack of availability of the obtained results, which is a significant limitation. Please delete the word "precise" in line 78 of the manuscript.